# The Impact of Digitization on the Formation of a New Model for Geospatial Data

Marina Jovanovic-Milenkovic [1,*] and Filip Petrovic [2]

1 Project Management College, Educons University, 11000 Belgrade, Serbia
2 Faculty of Architecture, University of Belgrade, 11000 Belgrade, Serbia; sevenarh@gmail.com
* Correspondence: marina.jovanovic.milenkovic@pmc.edu.rs

**Abstract:** The introduction of digitization has changed all spheres of business on a global level, including geospatial data. The general goal of the paper includes the formation of a new model of geospatial data management. The authors propose the formation of an eSpace model that includes the ePlan system. In order to achieve the goal of the paper, the authors conducted a survey in which representatives of local self-governments and holders of public authority participated through a structured online survey. A pilot study for the formation of a geospatial data model is an overview of spatial and urban planning. The focus is on looking at the real state of spatial and urban planning documents and the possibility of establishing a central database of spatial planning documents in digital format and its further distribution through a single system. In this way, easy access to digital plan data expands the community of users and enables communication with different groups of stakeholders. The introduction of the described model affects the further development of society as a whole.

**Keywords:** digitization; spatial and urban planning; eSpace; ePlan; geospatial data; data production; data distribution; value co-creation

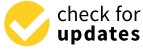



## 1. Introduction

How best to cope with the changes brought about by rapid urbanization and industrialization in order to maintain and promote human well-being is a current challenge for urban heritage conservation and management [1].

Global competitiveness in the 21st century is essentially based on improving business by employing information and communication technologies (ICT) [2]. The introduction of information and communication technologies promotes better work performance and greater efficiency and reduces costs related to all business segments. In this way, society's awareness of co-creation value is raised.

The subject of this paper is related to the digitization of geospatial data, with a focus on spatial and urban planning procedures, due to the fact that there is more and more talk about the implementation of smart cities and the introduction of smart devices in homes, as well as smart tools that are based on Bluetooth technologies [3].

City governments are absolutely required in the process of urban planning; utilizing their infrastructure and technologies, as well as cooperation from citizens, will be needed in order to approach the optimal smart city, because the ultimate goals of urban planning based on sustainable development are to improve the quality of life of citizens [4]. In order to fulfil the goal of a smart city, it is necessary to provide better access to high-quality and comparable urban planning data through a digital platform [5]. The use of ICT networks also needed for the future "smart city environment" [6]. In this regard, the process of urban planning, sustainable development, urban sprawl and urban form need to be focused around the concept of "smart growth" [7–9].

Based on the research literature, investing in smart cities affects the changing functions of urban planning and urban management [10]. Sucupira Furtado et al. (2023) examined

smart management and digital government. Their findings highlighted the significant role of digital data, transparency and commitment to justice among planners and policy makers to achieve a sustainable smart city [11]. Also, Mortaheb and Jankovski (2022) point out that a main role in improving the efficiency of urban services is played by GeoAI (geospatial artificial intelligence) [12].

These facts are confirmed by Jiang et al., who state that new business models developed as a result of new technologies affect the skill set and business practices among planning professionals [13].

Looking at such a significant part of the smart city, two main purposes of digitization in the field of geospatial and urban planning are identified:

- Enabling easy access to data from planning documents, with a high level of transparency.
- Creating national (regional/local) digital portal of homogeneous and high-quality geospatial data.

The authors certainly point to certain advantages related to the impact of digitization on the formation of a new model for geospatial data. Analyzing the previous literature, they mostly cite experiences using different software tools for digital visualization. In addition, they point to the importance of digital data exchange, which is why we need to work on improving the standardization of the database. It is also necessary to improve communication with interested parties [14]. One of the disadvantages of applying digitization is that local self-government units rely on electronics, which can make them unable to react or act in a scenario where these tools are not available.

The general goal of the paper includes the formation of a new model of geospatial data management, i.e., the ePlan as a future part of the eSpace for the digitization of planning documents.

eSpace is defined as part of the digital platform for the production of geospatial and nongeospatial data [15] of the National Infrastructure of Geospatial Data and the distribution of that data to citizens, the economy and public administration in the territory of the Republic of Serbia. The prerequisite for the formation of eSpace is the provision of high-quality and up-to-date spatial data and access to such data for their consistent use, in the processes of spatial planning and environmental protection, investment planning, construction, real estate valuation and real estate management [16].

An integral part of eSpace is ePlan. ePlan is a system for the production of planning documents and other legal regulations governing the use of space, including construction conditions [16].

The main hypothesis:

- It is necessary to establish a new model for geospatial data, an eSpace system, which includes a centralized electronic database of planning documents and all data related to land in Serbia, relevant for the use of space and construction.

More detailed hypotheses are as follows:

- Establishing the ePlan system, for the sake of an efficient production system of a centralized electronic database of planning documents and all data related to land in Serbia.
- Establishing an efficient distribution system of a centralized electronic database of planning documents and all data related to land in Serbia.

When formulating the hypotheses, the authors started from the necessity of presenting the real state of geospatial data as well as applying the principles of universal design in an integral way that is binding, checked by all stakeholders and that can be controlled. For the formation of eSpace, it is necessary to digitize data, create a system of production and then distribute the data via the platform. The main and specific hypotheses refer to urban and spatial planning; however, the focus can be shifted to other data of the urban environment, thus creating a new model of geospatial data management (Figure 1).

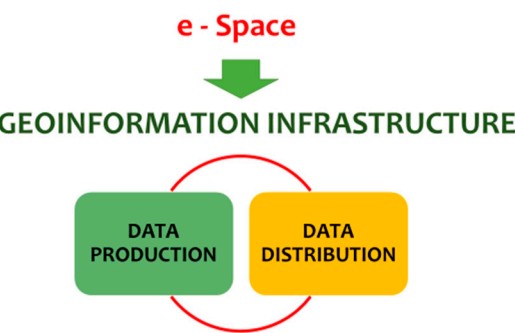

**Figure 1.** Espace and ePlan [16].

The article is structured as follows: In Section 2, we analyzed the existing models for the digitization of spatial urban plans in EU countries. In Section 3., using a survey as a data collection method, we examined representatives of local self-government units and public authority holders and analyzed their business processes. Section 4 presents the new model of the business process, the flow chart of the process and steps to form a new model for geospatial data. Finally, Section 5 presents the conclusions of the paper.

## 2. Research Background

Public administration and the services it offers, especially those of personal and civil status, have an important role in the lives of citizens. People turn to these services in order to solve various problems, and the interactions that people have with the representatives of the services, as well as the information circulated in the media, can influence public opinion about the performance and quality of these services [17]. These services have a technology layer that provides the required software and hardware infrastructures needed to provide digital services in smart cities [18]. This layer consists of infrastructures needed to collect, process, handle, and store urban data [19].

In this part of the paper, an overview of solutions related to spatial and urban planning and their digitization in the EU countries and in the countries that were part of the former Socialist Federal Republic of Yugoslavia is given.

EU countries follow development trends and keep up with the introduction of information and communication technologies in spatial and urban planning processes in order to increase the value of society. They started the process in phases, because they still use analog versions of the documents. The first phase that has been completed is the digitization of planning documents at the local level. The next phase involves planning at the national level [20].

According to the authors Nummi, Staffans and Helenius, digitization is expected to offer benefits in terms of land use planning and contributing to sustainability by enabling comprehensive plans and accurate assessments of plans. They state in their paper that in Finland, information model (IM)-based planning is initiated at the national level through development programs and laws. In 2020, an interoperable (i.e., based on a shared, machine-readable structure, syntax and semantics) IM was developed at the national level for local master and detailed plans. Various stakeholders were engaged in the development process, and feedback was gathered via a public web questionnaire [21].

The countries of the former Socialist Federal Republic of Yugoslavia have also recognized the benefits of digitizing these processes. Their analysis is significant because these countries, including the Republic of Serbia, have a common tradition in spatial and urban planning, as well as in competence to prescribe legal regulations on which land use and development is based. In the Republic of Serbia, planning still documentation exists, for the most part, in analog format, meaning the planning process is manual and paper-based. As an example, in the process of decision making for the development of infrastructure, there is a lack of data harmonization, which generates inaccurate and contestable results. For this reason, it is necessary to work on the digitization of planning documents. The described

requirements lead to the need to define and implement a new model for geospatial data, i.e., ePlan as part of the future eSpace.

In the system of the Republic of Serbia, and in accordance with the goals of the Government of Serbia, the Program for the Development of Electronic Government, and the Strategy for the Sustainability of Urban Development until 2030, the key challenge is to ensure the compatibility and comparability of digitized data from planning documents that were originally created in analog form [22,23]. Although the Government of the Republic of Serbia recognized the need for changes in urban planning, the active involvement of other interested parties is necessary. All target groups should be involved in the formation of a new model for geospatial data: professional associations of urban planners and architects, holders of public authority, local self-government units, civil society organizations, software manufacturers, etc., in order to ensure that digital planning documents meet the optimal standards required for their implementation. Given that the mentioned target groups have different competences and areas of action; all levels of public administration can benefit from the digitization of the planning system.

Easy access to digital plan data expands the user community and enables communication with different stakeholder groups. In this way, the inclusion of all target groups in planning processes is encouraged. Therefore, it becomes essential to consider the design and provision of service systems to manage geospatial data for spatial and urban planning and by changing the community's role from a passive user to an active co-creator of value.

Successful digitalization of planning documentation and data from that documentation implies defining standards and data models, establishing metadata and developing technical requirements for data in digital planning documentation at the national level [23]. In order to ensure the future use and continuous development of planning documentation, it is crucial to establish a comprehensive and uniform data structure [24].

This chapter provides an overview of the digitization of spatial and urban planning in the Republic of Slovenia and the Republic of North Macedonia.

### 2.1. Digitization of Spatial Planning in the Republic of Slovenia

Digitization of planning in the Republic of Slovenia was started with the ideas of experts in the field. This was followed by the support of state institutions, and the entire process started in 2017. The legal regulations adopted in the new Spatial Planning Law have been changed. It is prescribed that planning is carried out using the "Spatial Information System (SIS)", the creation of electronic databases of planning documents, as well as the application of data that are necessary for the use of space and construction [25].

The process itself also started in phases. The digitization process has five components [26]:

- Common infrastructure for spatial information;
- Spatial Information System (SIS);
- Information renewal of real estate records (land cadastre, building cadastre, etc.)—new IT system of the Geodetic Administration;
- Accepting new and improving existing data (graphical accuracy of the cadastre, scanning of archives, establishment of a database of built-up land—construction plots according to the principles of mass registration in the cadastre);
- Information and project management.

The Spatial Information System (SIS) in Slovenia consists of the following modules:

- Database;
- eConstruction database on the construction of buildings;
- ePlan—register of spatial plans and register of regimes;
- Data on built-up land (residential land, records of construction land).

Access to the modules is enabled through the central web portal, where all SIS users can find the necessary information and implement specific activities in specific procedures.

The ePlan module consists of [25]:

- An application for managing the procedure (workflow) during the development and adoption of plans, which supports the collection of data, conditions, etc., data exchange via web services, informing stakeholders, control mechanisms for the application of standards, and public participation. The limitation that exists is that the new plan cannot be adopted if it has not been prepared and adopted in this system.
- Register of spatial plans—GIS database.
- Register of legal regulations.
- GIS database, maintained by holders of public authority and owners of legal regulations.
- A system for monitoring the state of development in the field of urban planning.

It is the responsibility of the plan developer to review the standards that specify how spatial and urban plans are created and adopted, what they contain and how they are technically processed. The shortcoming of this system is that, from the legal perspective, all the entered data do not have a binding element for the holders of public authority [25].

### 2.2. Digitization of Spatial Planning in the Republic of North Macedonia

Digitization of spatial and urban planning in the Republic of North Macedonia implies a concept that is composed of several components [27]. Spatial information system consists of [27]:

- Real estate cadastre;
- Infrastructure cadastre;
- Central database of legal regimes (not established);
- Graphic register of construction land;
- Manual for creating spatial and urban plans—eUrbanism.

A division of labor was carried out in North Macedonia. The Spatial Planning Agency deals with the development of spatial plans. It also establishes and controls the central base of legal regulations, and as such is recognized as having the character of a state controller for all urban plans prepared by authorized architectural firms for the needs of local self-government units.

The graphic register of construction land is a unique electronic record, which contains all spatial and descriptive data for construction land. It implies the standardization of processes, procedures and data entry. This increases the transparency of the process of adopting urban plans, increases the percentage of realization of detailed urban plans, affects the increase in the number of investments for specific locations, a faster and simpler procedure for issuing a copy of the urban plan, and providing numerical/analytical data, etc. By introducing standards, mandatory layers and attributes are defined. It is the responsibility of the plan maker to comply with rules regardless of the choice of software used [27].

The advantages of the Graphic Register of construction land are [27]:

- Introduction of widely adopted technical GIS standards on the method of creating and storing urban plans in the central GIS database, which enables a transparent overview of locations where urban plans are in force, ownership of cadastral plots, construction land according to attributes, construction plots on state land, etc.;
- It is complementary with the "eUrbanism" system, which facilitates participation and the procedures for making, adopting and implementing plans.

The central base of legal regulations has not yet been implemented.

### 3. Research Approach

The effectiveness of planning implementation is the basis of the existence of planning legitimacy, and the issue of the effectiveness of planning implementation has been widely emphasized in both practical and academic circles [28].

Like any other system, public administration has certain principles that guide it and stand at the basis of its operation. Some of these principles [17,29,30] are:

- Legality—the authorities, its component institutions and the people working within it must act in accordance with the laws;
- Equality—people who use the services offered by the administration must be given equal treatment;
- Transparency—citizens must be allowed to participate in making administrative decisions, and have access to various information that may be of interest to them;
- Proportionality—there must be a balance between the needs of the people and the decisions taken by the administration, and it must analyze the impact that the measures taken have on the people;
- Impartiality—employees must act and solve problems objectively;
- Continuity—the administration must not interrupt its activity;
- Adaptability—people's needs are constantly changing, and the administration must keep up with them.

In the context of the local public administration, "the power of local administration is that it represents the common citizen" [31]. In this regard, citizens expect the public administration to be efficient, competent, modern and responsive to their problems [32]. This concept seems to ensure social value co-creation of sustainable social projects.

In the Republic of Serbia, the analysis of the state of spatial and urban planning indicates an excessive number of different types of planning documents. Competence for drawing up and adopting plans is divided between the Republic, the autonomous province, the city of Belgrade and local self-government units. Spatial and urban plans have a strategic and regulatory character in terms of enforceability.

In general, plan data include regulations for land use, and is produced by spatial planning authorities [33,34].

The spatial plan of special purpose areas and the spatial plan of local self-government units can be directly applied if, in addition to the strategic part, they contain elements of detailed regulation. The general urban plan contains a strategic plan without regulatory elaboration.

During the analysis, it was observed that the planning documents of the narrow area are not harmonized with the planning document of the wider area. Their harmonization is necessary, as well as, in the end, compliance with the Spatial Plan of the Republic of Serbia.

Priority planning documents for development and adoption, i.e., regional spatial plans and spatial plans of special purpose areas, are determined by the Spatial Plan of the Republic of Serbia.

The Law on Planning and Construction [35] establishes the deadline for the preparation of planning documents of spatial plans of local self-government units, general urban plans and general regulation plans that are under the jurisdiction of local self-governments. These plans also prescribe the mandatory preparation of detailed regulation plans. A detailed regulation plan can also be drawn up at the request of the investor, although its preparation is not prescribed by the wider area plan. Plans of a wider area prescribe a deadline in which planning documents need to be prepared. The creation of a spatial plan of a special purpose area or the creation of a detailed regulation plan can be initiated by the manager of a protected natural or cultural asset, the manager of the infrastructure, or an interested investor [31].

*3.1. Approach Method and Technique*

Considering the complexity of the described procedure as well as the necessity of introducing digitalization into this process, research was conducted in order to consider the possible implementation of the ePlan as a future part of eSpace. It implies looking at resources regarding the capacity of employees, adequate knowledge, and the use of ICT technologies.

The research is carried out by the survey method; the research technique is a questionnaire. The research was conducted between representatives of local self-government units

and representatives of holders of public authority at the republican and regional levels. It covered the period from 12 July to 19 July 2021.

Two research studies were conducted [31]:

- For representatives of local self-government units;
- For representatives of holders of public authority.

Representatives of local self-government units filled out a survey consisting of 26 questions. The total number of responses included in the sample for processing is 71, of which 15 are from cities and 56 are from municipalities. The sample was evenly distributed regionally with around 30% representatives from each region, with the exception of Belgrade, for which the questionnaire was filled out by the Belgrade City Administration (Figure 2).

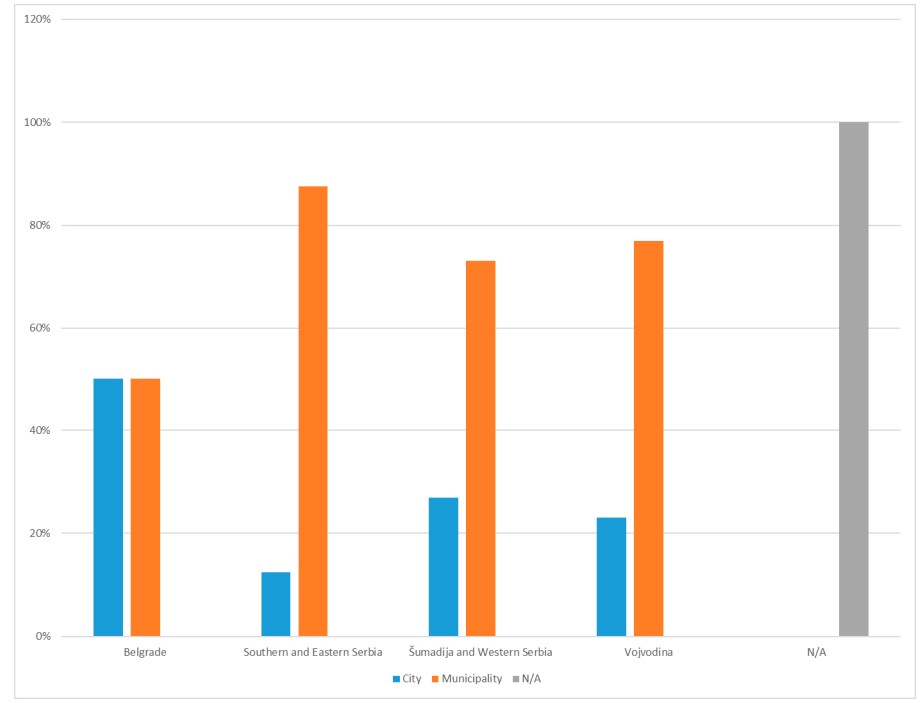

**Figure 2.** The total number of responses from local governments in the survey.

Representatives of public authorities filled out a survey consisting of 25 questions. The total number of responses included in the sample for processing is 62, of which 12 are at the national level, two are at the provincial level and 48 are at the local level (Table 1). The largest number of respondents have jurisdiction in the field of water supply, sewage and gas pipelines (42%), and electricity (16%). The largest number comes from Šumadija and Western Serbia (33%) and Vojvodina (26%) [31] (Table 2).

**Table 1.** Level of jurisdiction and region representatives of public authorities.

| Representatives of Public Authorities | Level | Region of Serbia % |
|---|---|---|
| The national level | 12 | |
| The provincial level | 2 | |
| The local level | 48 | |
| Šumadija and Western Serbia | | 33% |
| Vojvodina | | 26% |
| The other regions | | 41% |

**Table 2.** Percentage of jurisdiction by area.

| Area | % |
| --- | --- |
| Water, gas, sewage | 42% |
| Electricity | 16% |
| Heating | 8% |
| Environment | 6% |
| Traffic/roads | 6% |
| Cleanliness | 5% |
| Sanitary conditions | 2% |
| Cable network | 3% |
| Railway | 2% |
| Greenery | 2% |
| N/A | 2% |
| Forests | 2% |
| Waterways | 2% |
| Ski resorts | 2% |
| In total | 100% |

*3.2. Analysis of Research Results*

After conducting a survey with representatives of local self-government units and holders of public authority, an analysis of the responses was made. The analysis indicates that there are valid spatial plans at all levels in the Republic of Serbia. The Spatial Plan of the Republic of Serbia is legally valid. However, the lack of urban plans is evident. All cities have adopted general urban plans, but not general and detailed regulation plans. General regulation plans, as a new type of planning documents, were introduced into planning practice in 2009 but were not adopted within the legally prescribed deadline.

In practice, when there is no adopted planning document, general rules for arrangement and construction are rarely applied. It happens that location conditions are issued on the basis of a plan of a wider area that contains regulations, in the case when a detailed regulation plan has not been adopted within the prescribed period. Considering this practice, the general regulation plan, as a planning document that should be the basic urban plan and that should be directly implemented, did not meet expectations in practice. Because of all the above, it is necessary to seriously reconsider justifying the existence of different types of urban plans.

The main results of the surveyed representatives of local self-government units are that [31]:

- Most local self-governments always announce the start date of work on spatial and urban plans.
- Twenty-five percent of local self-governments announce the start date of work on the general urban plan, while the rest did not prepare this type of plans.
- In the process of drafting planning documents with the delivery of conditions, holders of public authorities at the republic level are most often late, and this is less often the case at the local level.
- Fifty percent of local self-governments organize a consultative process in the process of drafting planning documents.
- The consultative process is most often organized through the collection of proposals through the website of local governments, holding round tables or focus groups.
- The average number of comments per plan, during public inspection, is generally less than 10.
- Challenges for digitization are seen by most local governments in the insufficient number of personnel or in their inadequate training. Some of them state that they are aware that these shortcomings can be eliminated by educating the joint services of several local governments, while others suggest that it is necessary to find a way to hire additional quality staff.

- Commissions for plans of local self-governments generally meet once a month or as needed.
- Electronic sessions of the Commission for Local Self-Government Plans are rarely organized, although in most cases (63%) they are foreseen by the Rules of Procedure of the Commission, and the reason given is the insufficient equipment of local self-governments, which often do not have multimedia rooms, licensed software for online meetings or good internet.
- Most representatives of local self-governments enter data into the General regulation plan in CAD format (Figure 3). More precisely, on the basis of the answers received, the percentage ratio of the entry of plans in CAD format can be observed:
  - 48% of local self-government units enter data into the General Regulation plan;
  - 46% of local self-government units enter data into the Detailed regulation plan;
  - 34% of local self-government units enter data into the Space plan;
  - 29% of local self-government units enter data into the Urban project.

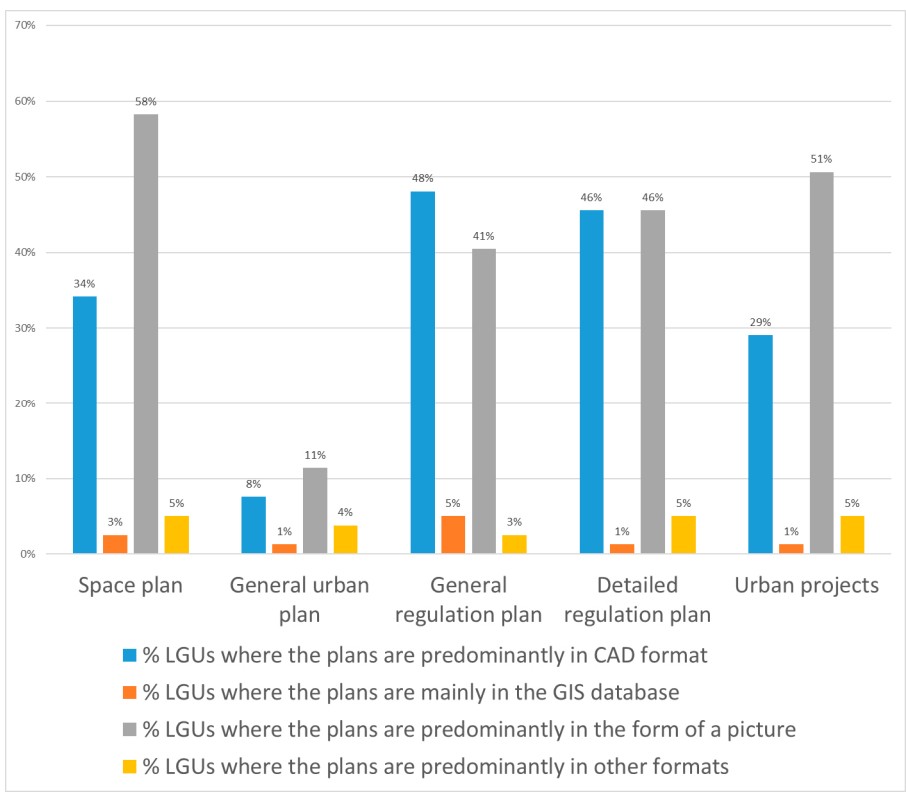

**Figure 3.** Predominant form of valid plans by type of plan (% local self-government).

The main results of the surveyed representatives of public authorities are that [31]:

- The largest number of holders of public authority derive their authority from the Law on Planning and Construction and the regulations adopted on the basis of that law, but also from sectoral laws such as the Law on Energy, the Law on Water, the Law on Communal Activities, etc., as well as from the acts of LGUs.
- Ninety-five percent of holders of public authority participate in the process of drafting planning documents by giving conditions, although other ways are mentioned, such as providing information, giving consent or opinions.
- Seventy-nine percent of holders of public authority state that they always submit documentation within the deadline (most of them within 6–15 days) (Figure 4).
- The majority of holders of public authority do not cooperate with other authorities or holders in the process of drawing up conditions, which enables them to act within the deadlines because they do not depend on the promptness of others.

- The majority of holders of public authority submit conditions (substances, consents, information) from their jurisdiction in analog form, i.e., in paper form, and less than 5% in CAD format or GIS database (Figure 5). More precisely, the percentage of data that are in paper format, and refer to the deadline for submitting documentation, are as follows:
  - ○ 56% of the data refer to Conditions;
  - ○ 34% of the data refer to Substrates;
  - ○ 52% of the data refer to Agreements;
  - ○ 53% of data on Information.
- No representative of the holder of public authority stated that some regulation, which governs their operations and competences, is a potential problem for the digitization of the process of creating spatial and urban plans.
- Sixty-five percent of holders of public authority believe that they have personal resources of adequate education (Figure 6).
- In 76% of cases, in the process of creating or reading documentation, representatives of public authority holders use Microsoft Office tools, but also other open and free software (Libre Office 7.2.4.1, Open Office Writer 4.1.11, etc.) (Figure 7). According to the answers received, the system software used by the employees participating in the planning documentation has the following structure:
  - ○ 76% of representatives know how to use the Central Record of the Unified Procedure;
  - ○ 2% of the representatives know how to use the service mode of the organ;
  - ○ 24% of representatives know how to use GIS tools;
  - ○ 50% of representatives know how to use CAD tools;
  - ○ 6% of representatives know how to use the Central Register of planning documents.
- Software used by holders of public authorizations to create documentation:
  - ○ CAD tools v.24.1 tools—52%;
  - ○ Microsoft Office 2019—76%;
  - ○ GIS 10.8 tools 23%.
- Software used by holders of public authority to read documents of other bodies:
  - ○ CAD tools—60%;
  - ○ Microsoft Office—77%;
  - ○ GIS tools 21%.
- The majority of holders of public authority find the challenges of digitization of their actions in human and technical capacities (on average, each service has four engaged employees who work on the creation or reading of documentation related to spatial plans) (Figure 8).
- A large number of holders of public authorities point out that the age of their equipment is the main problem for the digitization of the procedure (on average, computers are 6 years old).
- Eighty-two percent of public authority holders never engage external experts in the development of conditions for spatial and urban plans.

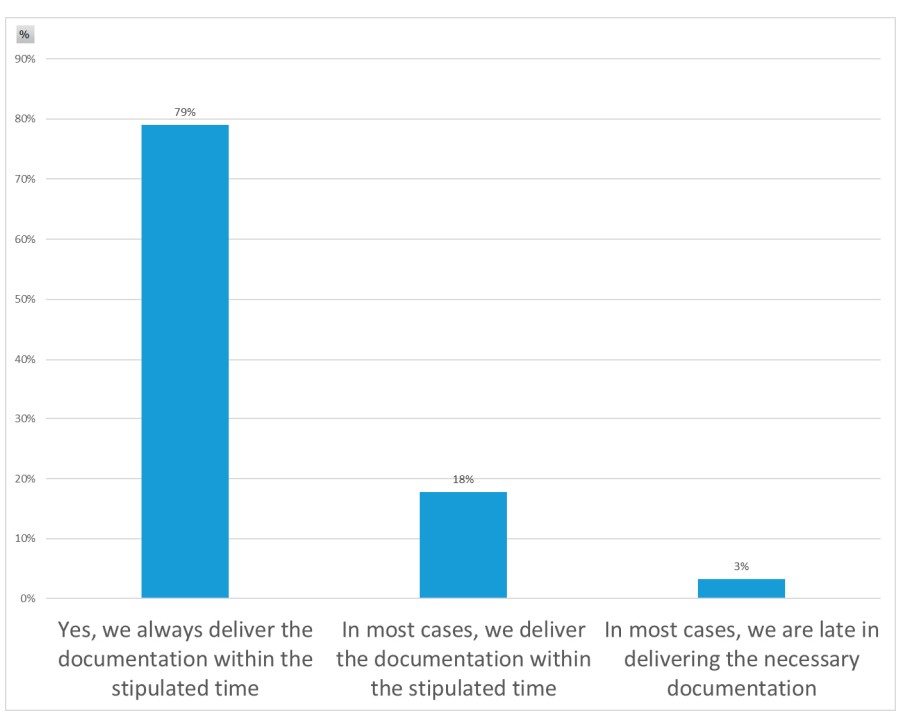

**Figure 4.** Timeliness of submission of documentation.

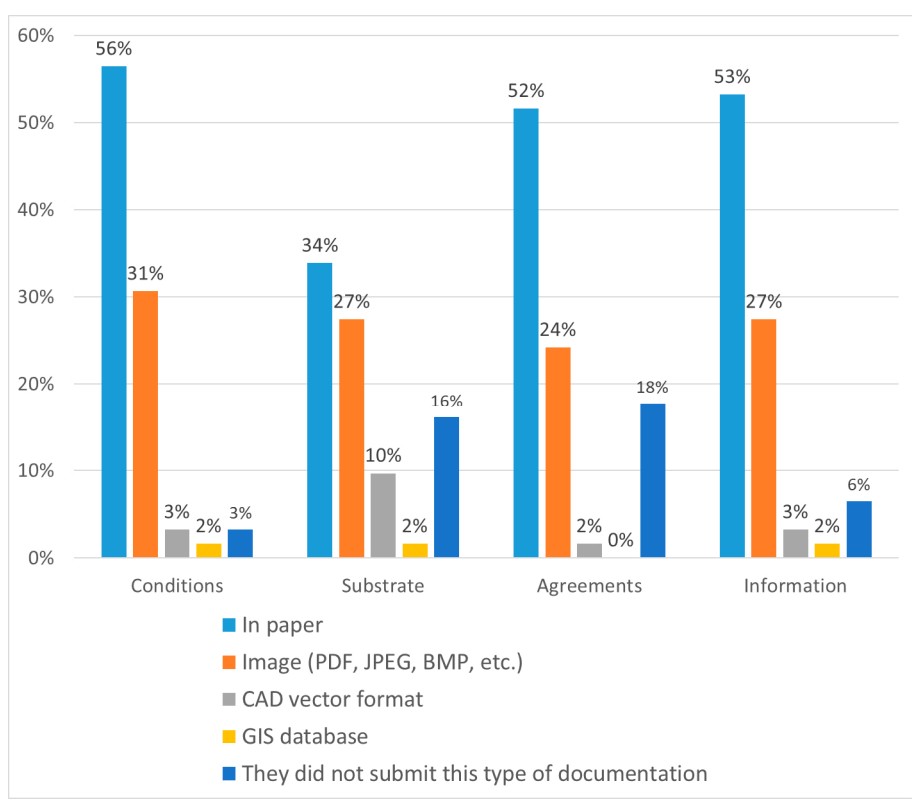

**Figure 5.** % of holders of public authorizations according to the average deadline for submission of documentation (by type of documentation).

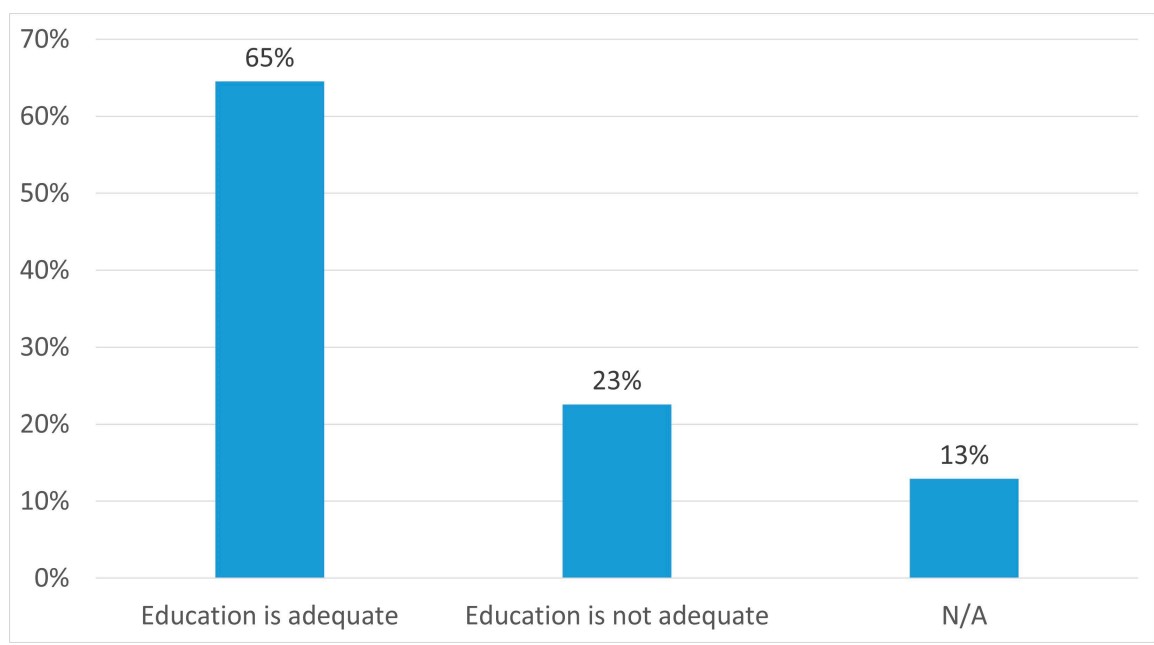

**Figure 6.** % of representatives of public authority holders—assessment of the adequacy of the education of engaged employees.

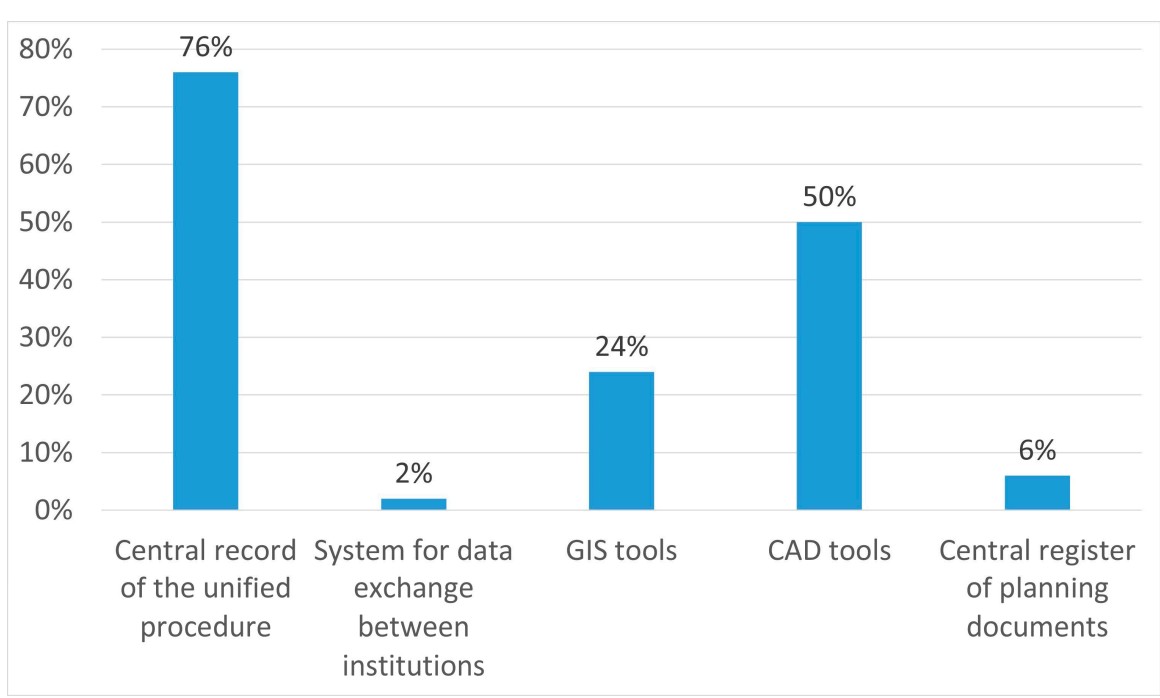

**Figure 7.** % of tools used for the publishing of documentation prepared by the holders of public authority.

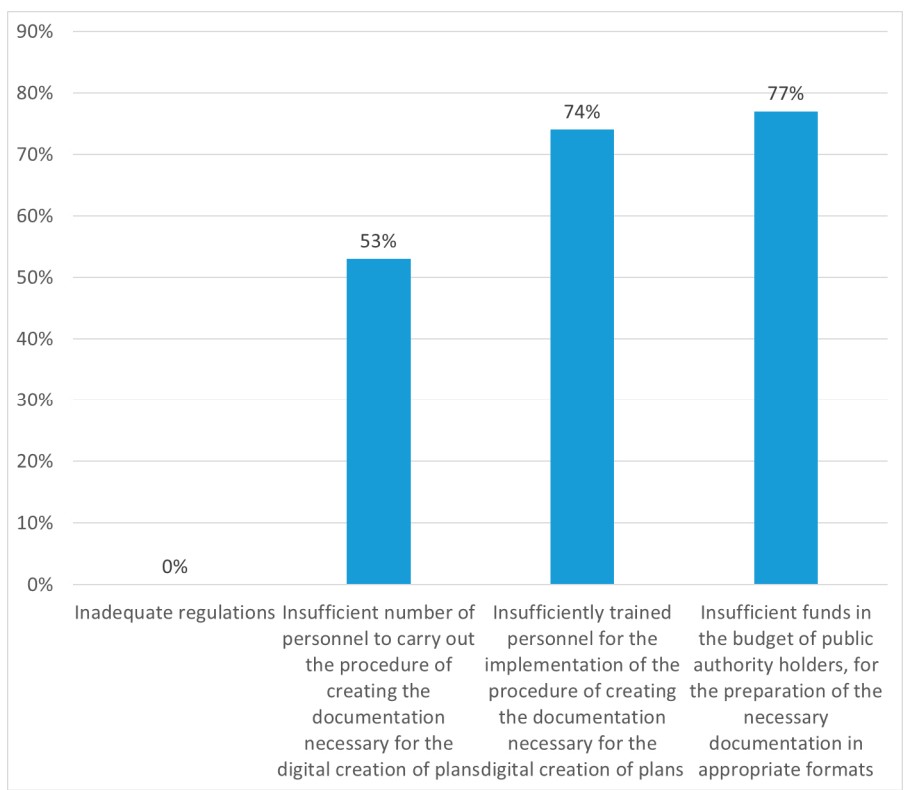

**Figure 8.** % of holders of public powers according to potential obstacles for digitalization of processes.

### 3.3. Analysis of the Legal Framework on Data on Planning Documents

In order to establish an efficient system of spatial and urban planning, it is necessary to establish a comprehensive public electronic database of all valid planning documents, as well as a unified electronic procedure in which these documents will be adopted. The analysis of the planning documents included, first of all, a detailed analysis of regulations related to planning and construction.

Considering the legal regulations and received responses from representatives of local self-government units and holders of public authority, it is necessary to amend the Law on Planning and Construction, in accordance with the application of ICT technologies [31]:

- Prescribe the establishment of a central database of all valid planning documents, which ensures their continuous public availability.
- Through the ePlan information system, as part of the unification of the electronic procedure, it is necessary to obtain the approvals prescribed by the Law on planning documents, as well as control procedures by the commission that controls the compliance of planning documents.
- Through the ePlan information system, prescribe that local planning documents and conditions in the area are monitored.
- The obligation to implement a unified procedure through the ePlan information system, in the process of adopting all valid planning documents.
- The unified procedure for the adoption of spatial and urban plans is fully implemented electronically, through the ePlan information system.
- The unified procedure includes plan makers, as well as all holders of public authority who are involved in that procedure, with deadlines for action appropriate to the electronic procedure.
- To communicate in the unified procedure exclusively through the exchange of electronic documents and submissions, except in the case of a special attachment that refers to special measures of organization and preparation of the territory for the needs of the country's defense.

- Prescribe the level of authentication required to access the ePlan information system.
- Prescribe that the data, i.e., planning documents registered in the central database of planning documents, produce legal effects towards conscientious third parties, which would give that database the status of a register.
- Establish a central base of legal regimes of holders of public authority.
- Prescribe the obligation of all holders of public authority to register their legal regimes in the central database of legal regimes.
- Establish a mechanism that will ensure the completeness of the central database of legal regulations and its compliance with the law and valid planning documentation, and in particular ensure that a sufficient volume of data are entered into the central database of legal regulations, so that location conditions can be issued based on them without additional inquiries.

## 4. New Model for Geospatial Data

### 4.1. Model of Business Process

Based on the analysis, the authors propose a model that will include the data of local self-government and public authorities. It implies the application of ICT technologies and digitization of the described data. Certainly, in order for the model to be applied, there must be adequate ICT equipment as well as a social component, which implies adequate user knowledge, user training, and communication and cooperation between users. The model implies three main business processes (Figure 9):

1. Data production;
2. Distribution database;
3. Distribution of data.

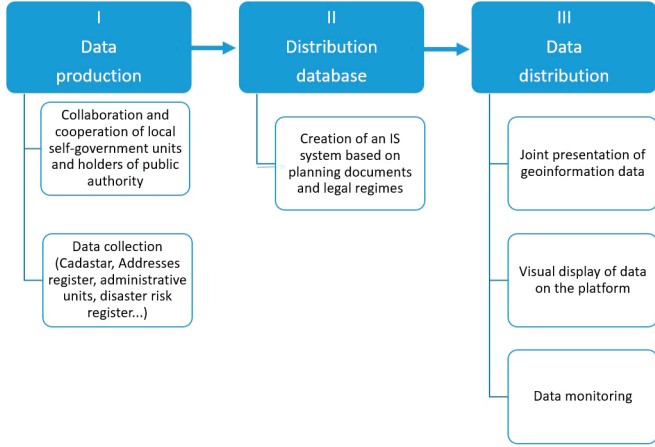

**Figure 9.** The three main business processes.

1. The Data Production process involves the collection of data from local self-governments (LSG) and holders of public authorizations related to a number of attributes:

    - Collection of urban and spatial plans managed by LSG, in the appropriate format'
    - Collection of data from the Ecadaster database managed by RGA (data on cadastral municipality, cadastral plot, area, legal property data);
    - Collection of data from the Address Register database managed by RGA;
    - Collection of data from the database for granting building permits managed by the Serbian Business Registers Agency;
    - Collection of data from the Disaster Risk Register on geographic location, data on hydrographic features, data on relief, data on meteorological-climatic features, data on demographic features, data on agriculture, data on material and cultural assets and protected natural assets, and data on plant cover in order to recognize

the risks of floods, landslides, forest fires, earthquakes, epidemics and pandemics and other risks;

- Other data from relevant databases.

For this process, it is necessary to submit a data format that is harmonized and standardized. For example, if some plans are made in another format, it is necessary to use techniques for data transformation from one format to another (such as the Feature Manipulation Engine—FME). Professional associations of urban planners and architects, holders of public authority, local self-government units, civil society organizations and other important participants participate in the described phase of the process.

2. The distribution database implies the creation of the ePlan IS system, which contains a unique central database that will enable:

- Establishing a central database of planning documents;
- Ensuring that spatial and urban plans are adopted through the IS system ePlan;
- Establishing a central base of legal regimes.

3. Data distribution implies a joint presentation of the situation in space, data monitoring, as well as joint geoinformation distribution components. In this way, we form the eSpace database. The publication of all geospatial data must be performed exclusively through the central eSpace portal.

### 4.2. The Flow Chart of the Process

Based on the three main business processes, a flowchart is formed. The first step is collaboration and cooperation between stakeholders. This confirms that everyone will collect data from their jurisdiction and ensure that the data are transparent and accessible to everyone.

Data collection implies that data are collected in an adequate format according to agreed standards. The collected data are entered in the register. Entry, change and deletion of data, entry of documents are registered in the register. We distinguish between two types of register: register of legal regimes and register of urban and spatial planning.

After entering the data into the Registers, the Data Distribution phase occurs, with the possible visualization of that data on a digital platform. In this way, a model for managing geospatial data is formed and additional value is created.

Based on the above, a new model is being formed that has its own benefits:

- All data are in one place;
- The data are digitized and easy access is enabled;
- Data are visible and transparent;
- All stakeholders communicate with each other and share their data.

The proposed flow chart of the process is shown in Figure 10.

In this way, the authors point to the benefits of forming a new model of geospatial data because its formation indicates the possible electronic display of all data located in real space. Based on this, it is possible to report and a standardized visual display containing all the data found in the database.

The entire model process introduces the concept of collaboration and co-creation, based on the recognition of the importance of accessibility and the principle of universal design for all people, shifting the focus from urban and spatial planning to geospatial data useful for all users. This indicates the necessity of an integral and comprehensive approach to the described problem and building a digital platform of geospatial data with a unique database, i.e., eSpace.

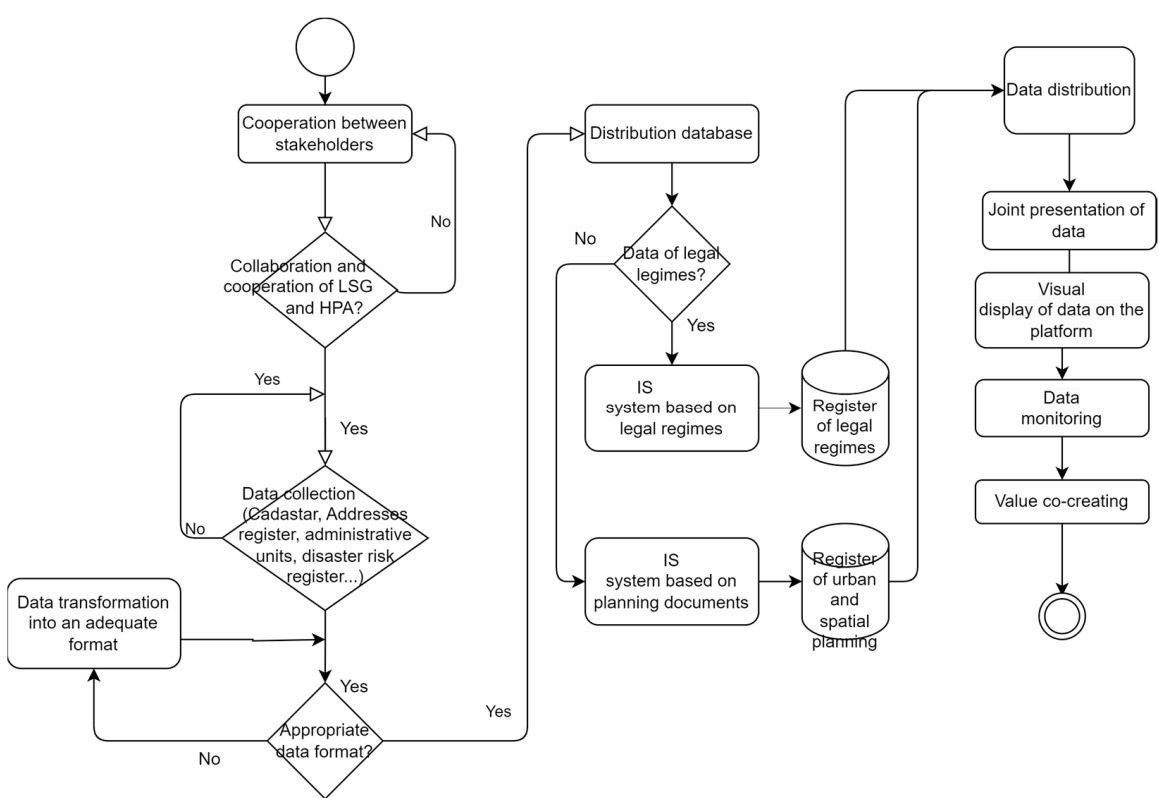

**Figure 10.** The flow chart of the process.

### 4.3. Steps to Form a New Model for Geospatial Data

As the digital economy expands, digitalization has become a significant global trend [36–38]. A rising number of enterprises has been embracing digital transformation with edge-cutting technologies to respond to this trend [39]. The digital transformation, as a process of data collection, storage and analysis using advanced digital technologies, has become a strategic selection for many firms to improve productivity [40,41]. As a result of the deep adjustments in technology and the market environment brought about by the advent of the digital economy, more and more firms are turning to digital technologies to encourage organizational optimization and speed up the pace of innovation in products and services. Value co-creation denotes that this process occurs through interaction between an organization and citizens. Thus, the concept of social value seeks to maximize the additional benefits created by the commissioning of projects and services, above and beyond the benefits of the products and services themselves.

In relation to the goal, paper hypotheses and analysis of the current state of spatial and urban planning in the Republic of Serbia, the conclusion is that it is necessary to create a new model for geospatial data, i.e., ePlan as part of eSpace (Figure 11).

The paper started from the special goals adopted by the Government of the Republic of Serbia through the "Program for the Development of Electronic Government in the Republic of Serbia for the Period from 2020 to 2022 with the Action Plan for its Implementation", which represent the basis of the implementation of digitized planning documents [22].

The "Strategy of Sustainable Urban Development of the Republic of Serbia until 2030" [23] supports digitization and the introduction of electronic services in the management of urban development, which also represents the basis for the development of the described digitization. This confirms the general goal of the paper.

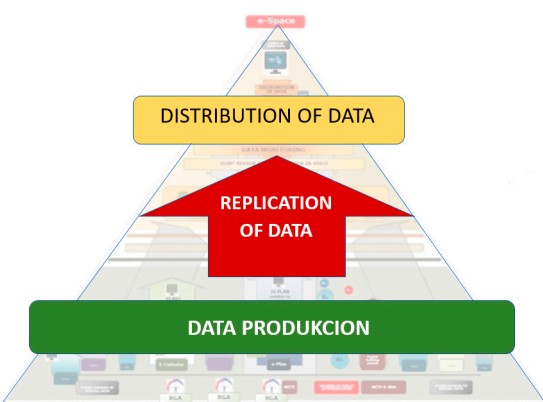

**Figure 11.** A system of production and distribution of a unique database of planning documents [31].

The main and special hypotheses were confirmed in terms of the necessary establishment of a system of production and distribution of a centralized database of planning documents. This activity develops and establishes a new model of geospatial data, an eSpace system related to the region of the Republic of Serbia.

Based on the research of the paper of local self-government and holders of public authority, and the obtained analysis, it is concluded that it is necessary to improve legal acts and enable their digitization in order to directly implement planning documents. The adoption of social value and value co-creation concepts is an attempt to introduce multidisciplinary and system views to sustainable management of geospatial data in spatial and urban planning.

Table 3 shows what needs to be achieved in local self-government units and with public authority holders in order to create a new model for geospatial data:

**Table 3.** What needs to be achieved for the new model in the local self-government unit and with public authority holders.

| The Following Needs to Be Done | Local Self-Government Units | Public Authority Holders |
| --- | --- | --- |
| 1. | Review and reduce the number of types of planning documents; | Digitize the process of obtaining conditions in the process of creating planning documentation |
| 2. | Digitize the entire process of preparation, coordination and monitoring of the preparation of planning documents | Establish rules on the use of available formats for all participants in the procedure |
| 3. | Centralize the procedure for adopting planning documents for local self-government units that have limited capacity | Implement a phased introduction of digitization, depending on the provided staff, equipment and financial resources |
| 4. | Acquire the IT equipment necessary for the digitization of the procedure for adopting planning documents | Review and take into account the current situation for registration in the cadastre of underground lines |
| 5. | Conduct regular employee training for the use of the information system | Make sure that copies of the plan and other documentation is sent in .dwg format, if the plot is georeferenced |
| 6. | Network the participants in the process of preparation, coordination and monitoring of the planning documents. | Send documentation in editable digital form, and spatial data in .shp format, through geoweb service. |
| 7. | Digitize the entire procedure, standardization and formatting of documentation (text and graphics) for the purposes of issuing conditions with up-to-date accompanying data from the real estate cadastre in digital format. | Work on the creation of a unified cadastre of underground utility installations to which all holders of public authority would have access. |

The future steps to form a new model for geospatial data are [31]:

- Develop and implement a unique ePlan information system through which planning documents and other legal regulations governing the use of space will be produced;
- Establish an Internal Portal for professional users, as an entry point for all actors in the process of spatial and urban planning, as well as holders of public authority to regulate conditions for the use of space;
- Provide electronic communication between all participants in the system for complete interactivity;
- Standardize the content, methodology and format of planning documents;
- Provide a software application for the technical control of planning documents during their creation, as well as their entry into the central database of planning documents;
- Consolidate all legal acts in the same place with planning documentation;
- Improve the capacity of human resources of representatives of local self-government units to perform tasks in the field of urban planning, by establishing intermunicipal cooperation;
- Provide systemic and public support in the implementation of the concept;
- Establish the Central portal eSpace, which should ensure easy availability in one place of all relevant information related to the conditions of use of space in Serbia, including construction conditions;
- Establish an open data portal on land use conditions in Serbia (Cadastre of immovable property and utilities maintained by the Republic Geodetic Authority of the Republic of Serbia (RGA); Central records of the unified procedure for issuing building permits maintained by the Agency for Economic Registers; Register of immovable properties in public ownership maintained by the Republic property directorate of the Republic of Serbia, etc.).

The possible functioning of the eSpace system, the establishment of the ePlan system, the production and distribution of a centralized electronic database of planning documents and all data related to land in Serbia are shown in the following figure (Figure 12).

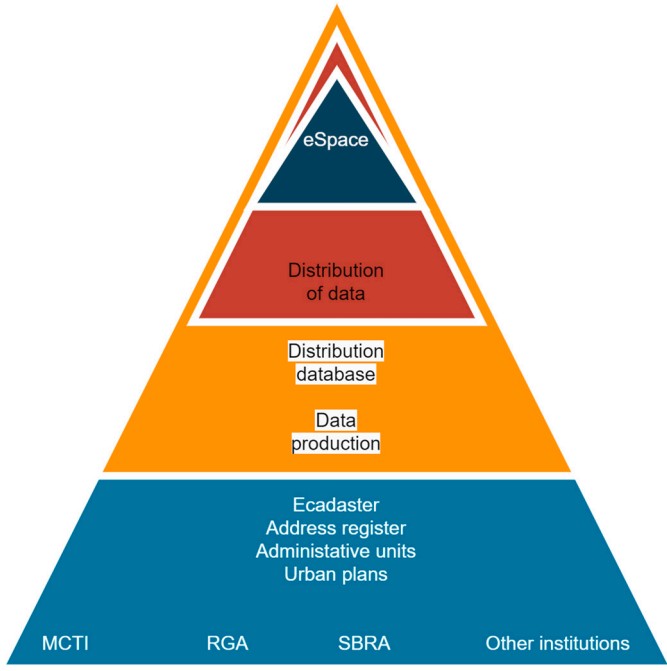

**Figure 12.** Graphic representation of the functioning of the eSpace system [31].

Based on this figure and in order to achieve the goal, the first step is the implementation of activities within pilot municipalities and with holders of public authority. The aim of the pilot activities is to check the applicability, adequacy and completeness of the set standards

and methodologies in practice, as well as their adaptation to the specifics and needs of the planning management of space and land in Serbia. In accordance with the results of the pilot activities, standards and methodologies would be approved, after which the existing legal framework should be amended and adapted to the needs of the eSpace system, as well as the preparation of technical specifications for the procurement of software, hardware and other resources necessary for the development of eSpace.

In light of this, urban planning becomes an essential tool that can help local governments define land-use policies to enhance local sustainable development [42–46]. Many cities enabled "open data" in order to enhance transparency and accountability to their citizens, but also to improve efficiency within the administration and promote local economic development (e.g., by facilitating new urban digital services using government data) [47]. Opening city data require technical expertise, breaking silos and vested interests both within and outside the administration, getting buy-in from data users and providers, bold political action and continuous action to carve out change in cultural mindsets [48,49].

Through research, the authors came to three groups of recommendations that should be implemented in order to form a geospatial data management model, i.e., eSpace. Given that it has been observed that the data are both in paper and electronic format, it is necessary to confirm the exact formats and data standards in order to make them acceptable for use in digital form. It was also noted that it is necessary to work on the introduction of software that enables the transformation of data from one format to another. Another important segment is the modification of legal documents in order to ensure a secure legal aspect by implementing a digital platform.

By collecting data, creating a register of urban and spatial planning and legal regimes, the authors indicate the possibility of data production and distribution as well as its visualization. The described workflow as a result of work contributes to the creation of a new model of geospatial data management.

The practical application of the results of the research carried out in the paper can be widely used in the field of geospatial data as:

- The basis for understanding the principles of universal design in order to achieve co-creation in practice through an integral procedure from design to monitoring.
- Incentives to empower all users to respond to the geospatial environment by active participation through the platform.
- The basis for the possible automation of the data processing process of various databases and the complete automation of the decision-making process on the accessibility of a certain urban environment.

By shifting the focus from urban and spatial planning to other databases and their registers such as cultural heritage, the health sector, and tourism of one region and others, the eSpace model for the complete management of geospatial data is formed. In this way, collaboration and co-creation is created as value creation for all users of the model, which further complements the importance of the model described in this way.

The main bearers of the described activities would be the institutions of the Republic of Serbia: the Ministry of Construction, Transport and Infrastructure (MCTI), the Republic Geodetic Authority (RGA), the Office of e-Government. It is also necessary to ensure the involvement of representatives of several institutions in the implementation of this reform, representatives of all ministries in whose jurisdiction the affairs of holders of public authority are, those obliged to introduce legal regimes, as well as representatives of other institutions. The entire process of communication would be based on the digital input of data for which institutions are responsible and their transparency between participants (Figure 13).

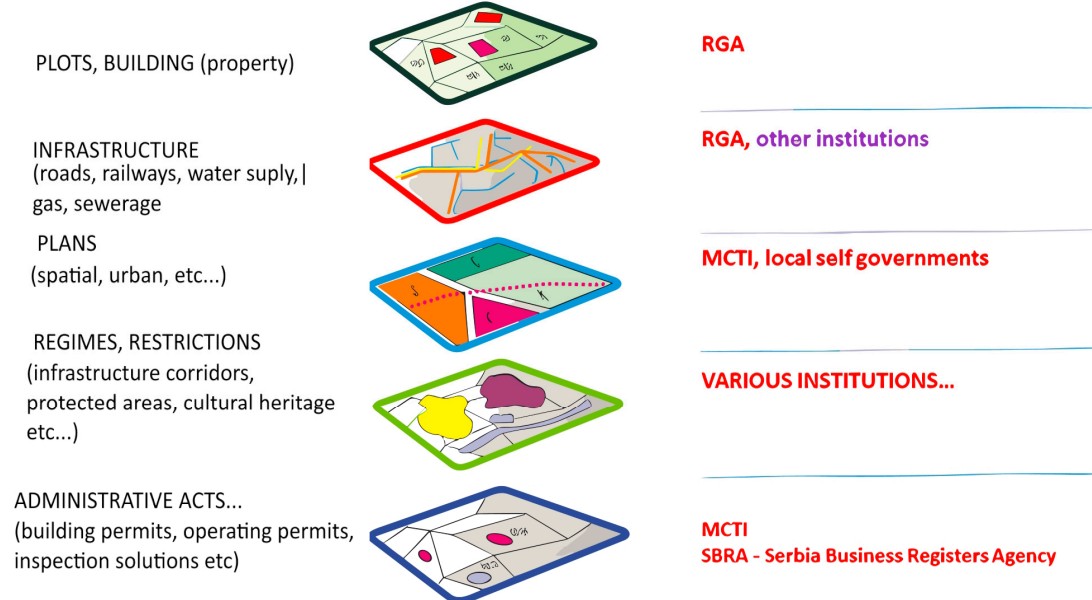

**Figure 13.** The digital input of data and their transparency between the participants [31].

By networking all data about each location, complete geolocation information is obtained. This creates the value of co-creation for public and private institutions and citizens of the country (Figure 14).

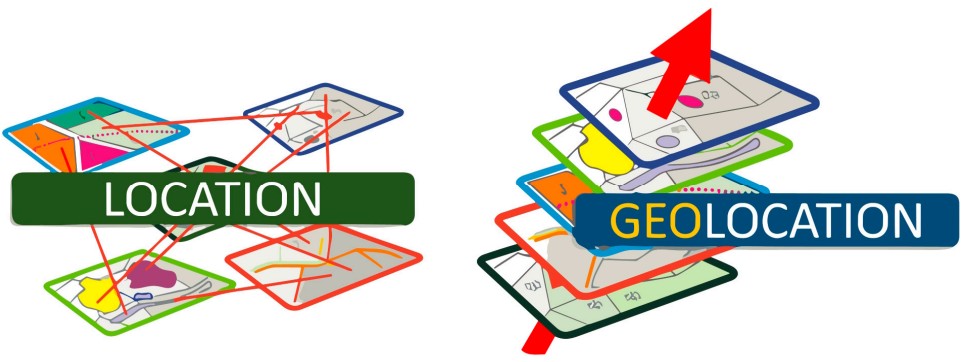

**Figure 14.** Graphic representation of connections with geolocations [31].

Keeping in mind internal and external human resources, there is a connection of several actors in one place: citizens, investors, local self-government units, state and other institutions. Between them, electronic communication is realized, which includes several models of communication: G2B, G2C, B2B, B2C and B2E. The two most common models of communication are G2C and G2B [5] (Figure 15):

- The Government-to-Consumer (G2C) model is used by potential investors and citizens to obtain information about locations. The G2C model is primarily concerned with obtaining information about the location by local governments.
- The Government-to-Business (G2B) model is represented in communication between local governments and other institutions. This communication includes information between individual legal entities. The basis of the G2B model concerns the provision of information, which aims to increase efficiency, reduce transaction costs and provide real-time information for all participants in the chain [38].

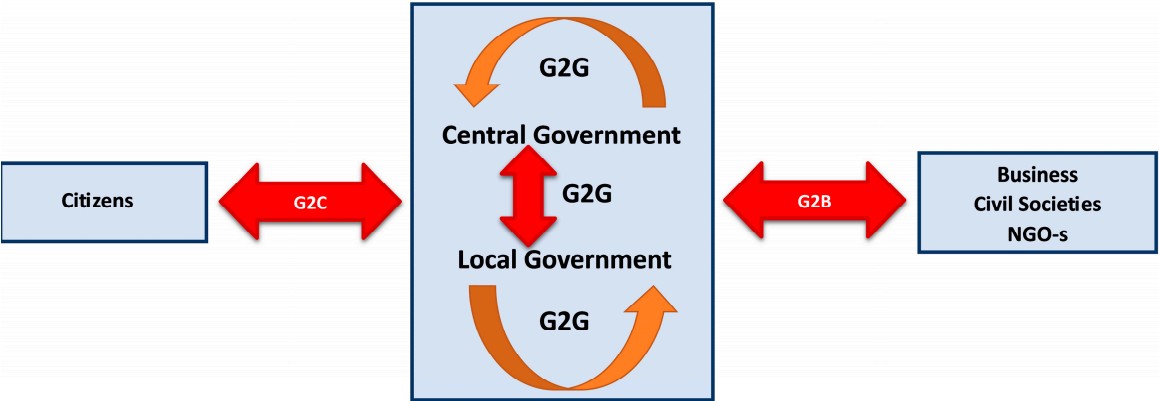

**Figure 15.** Electronic communication of participants [50].

## 5. Conclusions

Geospatial data management is vital to all business processes. By introducing digitization in this area, various interested parties and citizens are helped to better understand spatial phenomena, improve communication between interested parties [51] and support decision-making processes [52].

In the paper, the authors proposed a new model of geospatial data management, establishing the eSpace and ePlan systems, which is based on the joint creation of value in a sustainable project society. In accordance with the obtained results of the research, the necessity of implementing such a model is observed, where the concrete example in the Republic of Serbia shows that the incomplete Central Register of planning documents leads to the formation of eSpace, as a comprehensive centralized system, through which the transparent and efficient preparation and distribution of digitized planning documents and electronic data. This research adds the additional role of the interaction effect of government regulatory control procedures between political hazards and risk management [53]. Also, it is necessary to improve the current state of planning documentation and legal regulations related to the use and construction of land. In this way, the entire process tends towards the goal of efficient work of public administration and comprehensive and complete information to the public about the current state and possibilities of using the space. This is in line with the Strategic Action Plan for the National Spatial Data Infrastructure of the Republic of Serbia [54,55], which states that the Smart National Geospatial Data Infrastructure is based on values that greatly facilitate the access, sharing and use of geospatial data and services.

The stated goal of the paper is fully aligned with the general goal of the Government of Serbia [56], because it creates conditions for the implementation of all measures aimed at digitalization and the introduction of electronic services in the management of geospatial data, which confirms the necessity of implementing part of eSpace through spatial and urban planning. In this way, the management of these projects, which represent the largest investments in every society, will be more efficient, which will contribute to a faster and more significant creation of value for society [57]. On the basis of what is described in the paper, the proposed new model of geospatial data management indicates the advantages obtained through the application of digitalization.

However, the paper represents a pilot project that should be developed in other countries as well. The basis for this should be sought on the sustainability of projects together with social value, service system and co-creation of value.

**Author Contributions:** Conceptualization, M.J.-M. and F.P.; Methodology, M.J.-M. and F.P.; Formal Analysis. M.J.-M. and F.P.; Data analysis, M.J.-M. and F.P.; Writing—original draft preparation, M.J.-M. and F.P.; Writing—review and editing, M.J.-M. and F.P.; Supervision M.J.-M. All authors have read and agreed to the published version of the manuscript.

**Funding:** This research received no external funding.

**Institutional Review Board Statement:** Not applicable.

**Informed Consent Statement:** Not applicable.

**Data Availability Statement:** The data presented in this study are available on request from the corresponding author.

**Acknowledgments:** The authors would like to thank the Republic Geodetic Authority of the Republic of Serbia for the support provided in writing the scientific paper.

**Conflicts of Interest:** The authors declare no conflict of interest.

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
