# Peer review of "The Impact of Digitization on the Formation of a New Model for Geospatial Data"

_sustainability, doi:10.3390/su152216009_

Round 1
Reviewer 1 Report (Previous Reviewer 1)
Too much unnecessary information is included in the abstract. Please re-organize and make of point-by-point list of the main conclusions drawn by this paper
The introduction doesn't clearly present the motivation and novelty of the work.
Enrich the introduction part by adding recent studies related to your proposed work
Incorporate pros and cons of your study.
The authors need to better articulate the results and make sure that the figures and charts are properly illustrated with self-explanatory captions and labeling
The results are not explained properly, and need significant improvement in the writing
Too much unnecessary information is included in the conclusions. Please re-organize and make of point-by-point list of the main conclusions drawn by this paper
Incorporate future recommendations of your study.
Author Response
Dear Reviewer,
Thank you for the remarks you sent us. We have corrected the paper in accordance with your comments. The introduction, the obtained results and the conclusion have been improved in accordance with the goal and hypotheses given in the paper. The literature used is innovative. The paper was again technically arranged. More precisely:
Comments 1: Too much unnecessary information is included in the abstract. Please re-organize and make of point-by-point list of the main conclusions drawn by this paper
Response 1: We agree. The abstract has been modified in accordance with the written paper.
Comments 2: The introduction doesn't clearly present the motivation and novelty of the work.
Response 2: The introduction is more concretely written with a more precise aim of the paper
Comments 3: Enrich the introduction part by adding recent studies related to your proposed work
Response 3: Some considerations of other researchers who have published articles on a similar topic are included (lines 46-55).
Comments 4: Incorporate pros and cons of your study.
Response 4: We have described the advantages and disadvantages of the described model (lines 62-69). For better visibility, the structure of the paper is described (lines 102-107).
Comments 5: The authors need to better articulate the results and make sure that the figures and charts are properly illustrated with self-explanatory captions and labeling
Response 5: The obtained results and graphs were checked. Explanations are given in the paper. Also, new explanations were added in chapter 4.1
Comments 6: The results are not explained properly, and need significant improvement in the writing
Response 6: The analysis of the results is better described lines 363-369, 385-393, 403-420, 440-443.
Comments 7: Too much unnecessary information is included in the conclusions. Please re-organize and make of point-by-point list of the main conclusions drawn by this paper
Response 7: The conclusion is more concise and in accordance with the obtained paper results
Comments 8: Incorporate future recommendations of your study.
Response 8: Future recommendations are given in the conclusion. The researchers' literature of recent editions was added in accordance with the topic of the paper.
Once again, we thank you for your comments, because the paper is now certainly more specific, with better explanations.
Authors
Reviewer 2 Report (New Reviewer)
Dear authors.
I had the honor of meeting your article with the purpose of reviewing it.
The topic raised in the article is relevant and important. Moreover, the importance of this issue, as you rightly point out, extends not only to Serbia, but also to many other countries.
Let me, however, point out some shortcomings.
Purpose of the article
Not clear
In addition, expression (21-25)
“The goal of the paper indicates that it is necessary to raise the awareness of society, to introduce the concept of value co-creation, because conditions are created for the implementation of all measures aimed at digitization and management of electronic services in sustainable project society , at all countries.
This expression is formulated incorrectly.
conclusions
The conclusions of the article are not consistent with the purpose
This is due to the fact that the goal is not clearly expressed
Sources.
The extremely small availability of sources for the last three years, when your topic received incredible development in society, is immediately striking. I think I should look at more recent articles and statistical sources.
In addition, there are some stylistic remarks in the text that make it difficult to understand the message of the text.
So, for example, as I mentioned above, the sentence in the annotation that the goal indicates something is incorrectly composed
Also (lines 608-610)
The suggestion of the analysis of this paper is that the area of planning is ideal for inter-municipal cooperation and assignment of tasks from this competence to local self- government units with more significant resources.
It’s not clear which article you propose to analyze?
The general conclusion is that I believe that proofreading the article will make it consistent with the requirements of the journal and more convenient for the reader.
Good luck on your scientific journey
Author Response
Dear Reviewer,
Thank you for the remarks you sent us. We have corrected the paper in accordance with your comments. The introduction, the obtained results and the conclusion have been improved in accordance with the goal and hypotheses given in the paper. The literature used is innovative. The paper was again technically arranged. More precisely:
Comments 1: Purpose of the article -Not clear. In addition, expression (21-25): “The goal of the paper indicates that it is necessary to raise the awareness of society, to introduce the concept of value co-creation, because conditions are created for the implementation of all measures aimed at digitization and management of electronic services in sustainable project society , at all countries. This expression is formulated incorrectly.
Response 1: We have explained the goal of the paper in more detail in accordance with the hypotheses set in the paper. The sentences related to lines 21-25 have been corrected. The introduction of the paper was corrected. More recent observations by other authors in the field are given.
Comments 2: Conclusions. The conclusions of the article are not consistent with the purpose. This is due to the fact that the goal is not clearly expressed
Response 2: The conclusion has been improved and harmonized with the introductory part of the paper.
Comments 3: Sources. The extremely small availability of sources for the last three years, when your topic received incredible development in society, is immediately striking. I think I should look at more recent articles and statistical sources.
Response 3: In the introductory part of the paper, observations of other authors in accordance with the topic of the paper were added. The literature that we studied is a recent edition.
Comments 4: In addition, there are some stylistic remarks in the text that make it difficult to understand the message of the text. So, for example, as I mentioned above, the sentence in the annotation that the goal indicates something is incorrectly composed. Also (lines 608-610) - The suggestion of the analysis of this paper is that the area of planning is ideal for inter-municipal cooperation and assignment of tasks from this competence to local self- government units with more significant resources. - It’s not clear which article you propose to analyze?
Response 4: Lines 608-610 have been corrected. New explanations were added in chapter 4.1. The results of the paper are better described. The analysis of the results is better described lines 363-369, 385-393, 403-420, 440-443.
Comments 5: The general conclusion is that I believe that proofreading the article will make it consistent with the requirements of the journal and more convenient for the reader.
Response 5: The authors corrected the conclusion. It is better organized and more concise
Once again, we thank you for your comments, because the paper is now certainly more specific, with better explanations.
Authors
Round 2
Reviewer 1 Report (Previous Reviewer 1)
he authors answered correctly to all my comments.
Author Response
Dear Reviewer,
Thank you very much for the previously submitted comments.
We thank you for improving our paper based on that.
Authors
Reviewer 2 Report (New Reviewer)
The authors have significantly revised the text of the article.
It should be noted that the structure and text of the scientific article have improved.
However, there are still some shortcomings, the correction of which should improve this article
1. The first and most important thing in a scientific article is its purpose.
The purpose of the article remains not entirely clear. How can the purpose of an article be the construction and implementation of a model for geospatial data management (Lines 10-11)?
Maybe the goal is what you have written in the next sentence (lines 12-14)?
You are most likely proposing a new geospatial data management model
It is also not clear why the purpose of the work is indicated in the past tense (lines 70-73)
2. Regarding the text.
Some points seem redundant. For example, in forties 138-164 you describe the problem with the digitalization of documents in the Republic of Serbia. And you present this as a problem, but right there in lines 145-149 you write that this problem is not only recognized by the government of the republic, but that programs have been developed to correct this deficiency.
Author Response
Dear Reviewer,
Thank you for the comments you sent us. We have corrected the paper in accordance with your comments. More precisely:
Comments 1: The first and most important thing in a scientific article is its purpose. The purpose of the article remains not entirely clear. How can the purpose of an article be the construction and implementation of a model for geospatial data management (Lines 10-11)?
Maybe the goal is what you have written in the next sentence (lines 12-14)?
You are most likely proposing a new geospatial data management model. It is also not clear why the purpose of the work is indicated in the past tense (lines 70-73)
Response 1: We agree with the comment. The goal of the paper has been changed in accordance with your comment. Parts related to text lines 10-11, 12-14, i.e. 70-73 have been changed.
Comments 2: Some points seem redundant. For example, in forties 138-164 you describe the problem with the digitalization of documents in the Republic of Serbia. And you present this as a problem, but right there in lines 145-149 you write that this problem is not only recognized by the government of the republic, but that programs have been developed to correct this deficiency.
Response 2: We have added some clarifications in line with the comment. In part of the text, we explained the reasons why it is necessary for everyone to get involved in the formation of a new model. In addition to the Government of the Republic of Serbia, it is necessary to include other target groups in order for the entire process to be successful.
Authors
This manuscript is a resubmission of an earlier submission. The following is a list of the peer review reports and author responses from that submission.
Round 1
Reviewer 1 Report
The Results and discussion section is incomplete, since it should include at the end the considerations of the authors about the results obtained, he paper still lacks of a thorough discussion of the results and some real conclusions.
To this purpose, the reviewer suggests introducing a dedicated section just before the conclusion.
In the manuscript, the methods and techniques are well-known approaches and the repetition of the current literature. Hence the work does not bring a strong contribution to the respective field and thereby I have to suggest the rejection of the paper.
There are no figures and authors have introduced a discussion section. However, the paper still lacks a real discussion of results obtained and the conclusions drawn are still very weak.
Author Response
Dear Reviewer,
Thank you for the provided review regarding the paper. We accepted all comments and corrected the paper. More precisely, the following was corrected in the paper:
- The abstract has been corrected, it is linked to the title of the paper.
- The hypotheses of the paper are described in more detail in the Introduction paragraph. (lines 117-124)
- Section 3 with point 3.3 Results has been expanded. The authors described in more detail the proposed eSpace model related to the collection and distribution of data. In this way, a model for managing geospatial data using a digital platform is observed. The model is described with three main processes that are elaborated through steps. The authors also graphically presented a flow chart for the proposed approach.
- Section 4, which refers to the Discussion, was supplemented, the authors more clearly stated the contribution of the paper in accordance with the obtained research results, lines 643 -669. After the described recommendations by local self-government units and holders of public authority, it is described what exactly the new model contributes.
- In the Conclusion, an overview of the impact of digitization on the formation of new models and geospatial data is given.
- Also, the paper is supplemented with parts of the text that indicate the co-creation of the value of the model itself in the paper.
- Two new Figures were introduced, and Figures that were difficult to read were redone.
Best regards,
Authors
Reviewer 2 Report
In this paper, the authors proposed implementation of the ePlan as a future part of the eSpace for the digitization of plan-12 ning documents, by creating co-creating value. No changes found in the new version comparatively earlier version except references number.
1. In line 57, “In the Republic of Serbia, planning documentation exists still, for the most part, in analog format.” Please elaborate the format of analog. How planning documents store or maintain in analog format?
2. What is the format of digitation? Which data want to put in espace? Geospatial data word as well as palnning documents word both are vast. The authors should need to give clarification on the type of data, format of it, attributes, No of records, correlated values, etc
3. Which approach or methodology used to here to convert the data from a traditional approach to digitalization approach.
4. Authors mentioned only the Figures captions. But there are no figures.
5. In the related works, authors should need to discuss about the existed papers or works methodology or approach with their limitations, also include a summary table.
6. In section 3, authors need to include an algorithm or a step by step process for their proposed approach.
7. Suggested to include a flow chart to show the proposed approach process.
8. How could the authors analyze the collected data technically? What are the factors considered? Which mechanism or techniques or algorithm used to make the data as espace?
Needs to improve technically.
Author Response
Dear Reviewer,
Thank you for the provided review regarding the paper. We accepted all comments and corrected the paper. More precisely, the following was corrected in the paper:
- The abstract has been corrected, it is linked to the title of the paper.
- The hypotheses of the paper are described in more detail in the Introduction paragraph. (lines 117-124)
- In lines 57, the analogue format of planning documentation is explained.
- Section 3 with point 3.3 Results has been expanded. The authors described in more detail the proposed eSpace model related to the collection and distribution of data. In this way, a model for managing geospatial data using a digital platform is observed. The model is described with three main processes that are elaborated through steps. The authors also graphically presented a flow chart for the proposed approach.
- It was also explained that all data should be standardized and in a certain format. Data transformation from one format to another is possible using software such as FME.
- Section 4, which refers to the Discussion, was supplemented, the authors more clearly stated the contribution of the paper in accordance with the obtained research results, lines 643 -669. After the described recommendations by local self-government units and holders of public authority, it is described what exactly the new model contributes.
- In the Conclusion, an overview of the impact of digitization on the formation of new models and geospatial data is given.
- Also, the paper is supplemented with parts of the text that indicate the co-creation of the value of the model itself in the paper.
- Two new Figures were introduced, and Figures that were difficult to read were redone.
Best regards,
Authors
Reviewer 3 Report
Submitted paper has several significant problems:
1. It is hard to find connection between paper title and paper content. There is little in abstract or paper what is referring on formation of a "new model for geospatial data".
2. Mainly, paper presents statistical results of the conducted survey and lists long list of recommendations like in chapter 3.2 (pages 13/14) and chapter 4. (pages 15-17). For example in chapter 4. there are three groups of recommendations having 7, 11 and 24 listed recommendations (?!)
3. Synthesis of survey results is missing! Instead, new examples from other sources are introduced and conclusions made without link to survey results.
4. Some figures are unreadable or heavy readable and therefore useless (10, 11), for some is not clear why they are introduced and how they contribute to paper content (12), while for some connection between figure itself and figure description is not clear (1).
5. In conclusion there is nothing about formation of new model of geospatial data neither about impact of digitalization on specific model.
Missing the thorough analysis and scientific synthesis I find that this paper in way how it is structured and in a submitted form is not for publishing!
Even not being expert in English I find that paper should be improved considering English language (even Word is suggesting improvements). For example there are words missing in sentences, structure of sentences is strange, sentences are not well connected, etc.
Author Response

(The authors gave the same response as above.)

Round 2
Reviewer 1 Report
The authors addressed the comments satisfactorily. So, the manuscript can be considered for the possible publication
Author Response
Dear Reviewer,
Thank you for your comments. The paper is now certainly improved and clearer based on the provided comments of the reviewers.
Sincerely,
The authors
Reviewer 2 Report
The authors not updated the work as per the suggested comments. Especially, not considered the suggested comments on Methodology, techniques or algorithms and results analysis, etc.
Ok
Author Response
Dear Reviewer,
Thank you for your comments. We corrected the paper. The paper is now certainly more transparent compared to the original version. By reading the paper, other authors can see the goal of the paper and how we came to the results in the paper. We appreciate your comments.
In this comment, we will give an overview of the improved parts of the paper:
- By correcting certain parts of the paper, the paper got a clearer structure. The content of the paper is now more specific, and the goals and achievements in the paper are more clearly stated compared to the previous version of the paper. The hypotheses are described more precisely. The method and research techniques used in the paper are described in more detail. The title of the paper is better connected with the other Sections in the paper. We have changed the names in some Sections along with the contents of the Sections.
- The Abstract section has been corrected. It is more precise, now it gives a better explanation of the content of the paper itself.
- The Introduction section is supplemented with activities that are elaborated in the paper (lines 125-133). In this way, the flow of paper is indicated to the reader.
- Section 3 has an improved structure, the subheadings have been specified.
3.1. Approach method and technique
3.2. Analysis of research results
3.3. Analysis of the legal framework on data on planning documents
Section 3.1 has been corrected. The applied method and technique are described (lines 299-305). The research method is survey research, the research technique is a questionnaire. The target groups of the survey are: representatives of local self-government units and representatives of holders of public authority. The size of the sample in the research is adequate and provides complete information about the work, the capacity in the region.
- Section 4 has changed its name. We agree that it is not a real discussion, but recommendations and future steps for the formation of a new model. The new title is the New model for the geospatial data. Within this section, areas are divided:
4.1. Model of business process
4.2. The flow chart of the process
4.3. Steps to form a new model for geospatial data
Section 4.1. was supplemented. Three main processes are described, on which the new model for geospatial data is based. Also, a possible approach to converting the data from a traditional approach to digitalization approach is explained. In addition to the textual part, a graphic representation of the process is given.
Section 4.2. has been corrected with a more detailed explanation regarding the flow chart of the new model. Registers are created by submitting data in an adequate format. Data from the register are further distributed to interested parties. A graphic representation of the flow chart is given. By adding this section, a more complete representation of the results and the way of forming the model is given. Connect the model to the data flow.
Within Section 4.3. a new table has been added with the activities that are necessary for stakeholders to do. Future steps for the formation of a new geospatial data model have also been reduced.
- In the Conclusion section, the impact of digitization and the importance of forming a new model are explained
Sincerely,
The authors
Reviewer 3 Report
I am confused with response received from authors and changes introduced in the paper itself. The corrections made are rather introducing more confusion than clarifying the previous comments. This refers for sentence in lines 16-18, 24-29 in Abstract. Further improvements are trying to explain intention of the proposal (model) but all explanations remain on general level while then in Discussion we have 35 points of action which should be undertaken to develop proposed model. Reading this paper thoroughly one can conclude that conclusions and points of action are given and research - conducted survey should support those statements. It is simply not clear from where points of action in chapter Discussion (if this is discussion at all) are coming from the paper.
Therefore, I can not decide other than to reject this paper since improvements, in my understanding, did not respond on any of my previous comments.
As mentioned in 1st review it is my opinion that paper requires major improvement in English language.
Author Response

(The authors gave the same response as above.)
